# Metabolic Bone Disease in Children with Intestinal Failure and Long-Term Parenteral Nutrition: A Systematic Review

**DOI:** 10.3390/nu14050995

**Published:** 2022-02-26

**Authors:** Simona Gatti, Sara Quattrini, Alessandra Palpacelli, Giulia N. Catassi, Maria Elena Lionetti, Carlo Catassi

**Affiliations:** 1Department of Pediatrics, Marche Polytechnic University, 60123 Ancona, Italy; sara.quattrini29@gmail.com (S.Q.); ale_palpacelli@libero.it (A.P.); m.e.lionetti@univpm.it (M.E.L.); c.catassi@univpm.it (C.C.); 2Department of Pediatrics, University La Sapienza, 00185 Rome, Italy; giulia.catassi@gmail.com; 3Center for Celiac Research, Mass General Hospital for Children, Boston, MA 02114, USA

**Keywords:** intestinal failure, short bowel, bone disease, bone density, osteoporosis, DXA, vitamin D, parenteral nutrition, calcium balance

## Abstract

Metabolic bone disease (MBD) is a possible complication of intestinal failure (IF), with a multi-factorial pathogenesis. The reduction of bone density (BMD) may be radiologically evident before manifestation of clinical signs (bone pain, vertebral compression, and fractures). Diagnosis relies on dual-energy X-ray absorptiometry (DXA). Incidence and evolution of MBD are not homogeneously reported in children. The aim of this systematic review was to define the prevalence of MBD in IF children and to describe risk factors for its development. A comprehensive search of electronic bibliographic databases up to December 2021 was conducted. Randomized controlled trials; observational, cross-sectional, and retrospective studies; and case series published between 1970 and 2021 were included. Twenty observational studies (six case-control) were identified and mostly reported definitions of MBD based on DXA parameters. Although the prevalence and definition of MBD was largely heterogeneous, low BMD was found in up to 45% of IF children and correlated with age, growth failure, and specific IF etiologies. Data demonstrate that long-term follow-up with repeated DXA and calcium balance assessment is warranted in IF children even when PN dependence is resolved. Etiology and outcomes of MBD will be better defined by longitudinal prospective studies focused on prognosis and therapeutic perspectives.

## 1. Introduction

Intestinal failure (IF) is the consequence of several gastrointestinal conditions that determine the necessity of parenteral nutrition (PN) to maintain adequate growth and fluids/energy balance [1]. Short bowel syndrome (SBS) is the principal cause of IF in children; more rarely, IF is the result of a congenital enteropathy or a severe motility disorder. SBS generally follows a bowel resection in the neonatal period for a gastrointestinal anomaly (atresia, gastroschisis, and omphalocele) or an acquired event (neonatal volvulus or necrotizing enterocolitis). More rarely, SBS can develop in older children because of a surgical resection for several conditions, including inflammatory bowel disease (IBD), volvulus, and ischemic events [2]. The reversibility of IF and the achievement of enteral autonomy depend on the underlying condition and the surgical outcomes but are also related to both nutritional and medical treatments. PN has a crucial role in the different phases of IF management; in fact, PN represents the mainstay of treatment in the very early phase of SBS, a medium-term solution in reversible IF cases (when the process of intestinal adaptation progressively encompasses the bowel loss), and a life-long strategy in the most severe forms of IF. Advancement in home PN (HPN) and enteral nutrition (EN), both in surgical techniques and in medical strategies and particularly the development of specialized, multidisciplinary rehabilitation centers for the medical care of these vulnerable subjects, have notably improved the outcomes of IF children. Despite the progress in the field, children on HPN remain at risk of developing multiple serious complications, such as septic events, liver disease, renal dysfunction, and growth failure [3]. 

Metabolic bone disease (MBD) is another possible complication of IF. This term includes several conditions of alteration of skeletal homeostasis, including rickets, osteomalacia, and osteoporosis. Rickets is characterized by insufficient mineralization of the epiphyseal plates, leading to architectural changes and, possibly, skeletal deformities, such as chest bone softening, cranial bossing, bowing, and craniotabes. Osteomalacia or “soft bones syndrome” develops after the closure of the epiphyseal plates. Osteoporosis is characterized by low net bone mass and normal mineral-to-collagen ratio, leading to microarchitectural changes and resulting in increased fragility and enhanced risk of fracture [4]. MBD can occur in absence of clinical manifestation or present with bone pain (generally hip and back pain), vertebral compression, or long bone and vertebral fractures, along with other deformities. 

The pathogenesis of MBD in IF is multifactorial, including malabsorption of calcium, phosphorus and magnesium; vitamin D deficiency; reduced physical activity; an inflammatory underlying gastrointestinal condition (e.g., IBD); and toxicity related to some PN compounds [4]. Diagnosis of MBD relies mainly on radiological techniques, particularly on dual-energy X-ray absorptiometry (DXA), a very accurate, fast, and low-radiating method for assessment of bone mineral content and density [5]. The bone mineral content (BMC) refers to the bone content found in a specific bone area, and it is measured in grams, while bone mineral density (BMD) is the BMC divided the bone area, and it is expressed in grams/cm^2^. Bone mineral apparent density (BMAD) measures the bone density in relation to the bone volume (grams/cm^3^). DXA is suggested by the current guidelines as MBD screening in the follow-up of children on long term PN [6]. Serum markers of bone turnover provide adjunctive information on dynamic changes in bone metabolism, indicating nutritional deficiencies and the necessity of supplementation. 

While in adults dependent on PN, prevalence and risk factors of MBD have been clearly elucidated [7,8,9,10,11,12], the incidence of MBD in IF children has been sporadically reported. Furthermore, considering the vulnerability of IF children, it is possible that specific factors in this category of subjects contribute to the development of MBD. The aim of this systematic review was to investigate the prevalence of MBD (based on radiologic criteria) in IF children and to assess nutritional and other risk factors for MBD development and evolution. 

## 2. Materials and Methods

This systematic review was conducted following the guidelines of the Preferred Reporting Items for Systematic Reviews and Meta-Analysis (PRISMA) [13]. The protocol was registered on the PROSPERO international prospective register of systematic reviews: https://www.crd.york.ac.uk/prospero/, PROSPERO acknowledgement of receipt (300608) (accessed on 28 December 2021).

### 2.1. Eligibility Criteria

Population, Intervention, Comparator, Outcome, Study Design (PICOS) framework was used to determine the eligibility criteria [14]. We included studies that investigated prevalence and risk factors for MBD (defined with radiographic techniques) in children (0–18 years) with IF. Inclusion criteria were as follows: studies published between 1970 and 2021, English language, involving children (0–18 years) with IF and prolonged PN, and bone disease assessment with a radiological method (standard radiography, DXA, tomography, others). Randomized controlled trials (RCTs); observational, cross-sectional, and retrospective studies; and case series were included. Systematic reviews, meta-analysis, case reports, studies conducted in adults (>18 years), studies where the assessment of bone disease was based only on clinical, or serum markers or other non-radiological measures were excluded. 

### 2.2. Information Sources and Search Strategy

A systematic literature search was conducted through electronic bibliographic databases, in particular PubMed, Science Direct, Cochrane Library, and ERIC. The following keywords were used to include the more relevant studies: “parenteral nutrition” or “parenteral” AND “bone” or “bone health” or “osteoporosis” or “osteopenia” AND “intestinal failure” or “short bowel”. Moreover, references of the selected studies were screened to identify other relevant articles. The search was performed up to September 2021, and a last access was repeated in December 2021 in order not to exclude any updated study. 

### 2.3. Study Selection, Quality Assessment and Data Extraction

Titles were collected by two reviewers (S.G., A.P.) who removed duplicates. Abstracts and full articles were individually reviewed by three authors (S.G., A.P., and S.Q.). All full-text articles that satisfied eligible criteria underwent a quality assessment through the Academy of Nutrition and Dietetics Quality Criteria Checklist (QCC) for Primary Research. The QCC includes fourteen questions (four relevance questions that address applicability to practice and ten validity questions) based on the Agency for Healthcare Research and Quality (AHRQ) domains for research studies [15]. Each question can be answered with “yes”, “no”, “unclear”, or “not applicable (N/A)”, and a rating of positive, neutral, or negative can be assigned depending on the answers. Quality assessment and data collection process was independently performed by three reviewers (S.G., S.Q., and A.P.) who checked separately for accuracy and consistency of each selected study. An extraction form on a specific worksheet using the Microsoft Excel 2016 software was created and used to extract data. In particular, the following data were collected from the selected studies: study details (design, participants, country, period), IF- or PN-dependence definition, participants (number, age, gender, length of PN, prevalence of ongoing PN), bone disease definition (based on DXA or other radiologic techniques and clinical judgment, i.e., prevalence of fractures), DXA parameters (BMC in grams, BMD in grams/cm^2^, bone mineral apparent density (BMAD) in grams/cm^3^, and z-scores and/or adjusted z-scores), prevalence of bone disease and/or comparison with control group, serum levels of markers of calcium metabolism or prevalence of vitamin D insufficiency and/or deficiency (serum and urinary calcium and phosphorus, vitamin D, parathormone, alkaline phosphatases, other markers), risk factors (and type of analysis) correlated to MBD or to DXA parameters (i.e., age, type of IF, PN duration, quantity or composition, levels of bone markers and vitamin D), and follow-up results. Prevalence of MBD was estimated by summing all the patients with at least a DXA parameter ≤−2 z-score (at first DXA scan) divided by the total number of patients having a DXA scan in the same cohorts. If different studies clearly included patients from the same cohort, prevalence estimates were only registered once for each cohort.

## 3. Results

### 3.1. Study Selection

The systematic research identified a total of 650 articles. A total of articles were removed after title screening, and 114 abstracts were selected for eligibility. Seventy-three articles were full text screened and fifty-four excluded. One further article was included by reference snowballing. Twenty studies were included in the final review [16,17,18,19,20,21,22,23,24,25,26,27,28,29,30,31,32,33,34] (Figure 1).

All the selected papers underwent the QCC assessment and were considered suitable for data collection process and summary measures (Table 1). Table 2 summarizes the main characteristics of the included studies.

### 3.2. Study Design and Population

All studies were observational, and six included a control group (case-control studies) [18,19,20,24,25,34]. All except one were single-center studies: 7 from U.S. centers [16,18,19,22,26,27,28], 1 from Canada [25], and 11 from Europe [17,20,21,23,24,29,30,31,32,33,34], and 1 study reported data from 2 U.S. centers [24]. The 20 studies included patients from 15 different cohorts (6 studies from 3 cohorts [17,21,29,30,33,34]) and 3 studies from one cohort [19,22,24]). Eight studies (40%) reported longitudinal data [16,17,20,24,26,28,29,30]. 

The number of participants varied from 7 to 123, with a median number of 19 participants. In total, data from 640 IF subjects (351 males, 55%) were included in the review. Case-control studies compared data from 77 IF children with 131 (age- and gender-matched) controls [18,19,20,24,25,34]. Sixteen studies reported data from children [16,17,18,19,20,23,24,25,28,29,30,31,32,33,34], and two studies included children and young adults [21,22], while two studies did not specify the age of participants [26,27] although it was clearly indicated that data were collected from children with IF. 

There was considerable variation among definitions of intestinal failure and PN dependence, with minimum length of PN (to define the “IF condition”) ranging from 1 to 6 months. Eight studies exclusively included subjects on PN at the time of bone assessment [16,17,24,25,30,31,32,33], three studies focused on subjects after PN discontinuation (PN duration ranging from 1 to 78 months) [18,19,21], and nine studies included IF patients independently from being on PN [20,22,23,24,26,27,28,29,34]. In the eight studies reporting longitudinal data, median follow-up varied from 1 to 6.2 years [16,17,20,24,26,28,29,30]. 

### 3.3. Bone Health Parameters and Bone Disease Definitions

MBD was measured by DXA in 16 studies [19,20,21,22,23,24,25,26,27,29,30,31,32,33] and by single-photon absorptiometry by one study [18]. High-resolution peripheral quantitative computed tomography (HR-pQCT) was used in the most recent paper in comparison to DXA [34]. Three studies described bone disease and osteopenia based on X-ray [16,17,28]. Seventeen studies included data on biochemical bone disease markers [17,18,19,22,23,24,25,26,27,28,29,30,31,32,33,34], while occurrence of fractures was reported by eight studies [16,20,23,24,26,27,28,29]. 

Thirteen studies reported definitions of bone disease based on DXA parameters (compared to reference values extrapolated from healthy reference populations matched for age, ethnicity, and gender). Four studies defined MBD as one of the DXA parameters including BMC, BMD, and BMAD below −1 z-score [20,23,25,26]. Seven studies indicated as MBD one of the previous parameter inferiors to −2 z-score [22,24,27,29,30,33,34], while two studies considered both the cut-off values [31,32]. An adjustment of DXA parameters for bone characteristics (bone age or bone size) or for anthropometric measures (ideal weight or height or statural age) was considered by seven studies [22,23,25,29,30,33,34].

Vitamin D status was investigated by determination of serum concentration of 25-hydroxyvitamin D concentration (25-(OH)D) by most of the studies [17,18,19,22,23,24,25,26,27,28,29,30,31,32,33,34], while two studies also reported measurement of serum 1,25-dyhydroxyvitamin D (1,25-(OH)2D) [24,32]. A serum level of 25-(OH)D between 15 and 30 ng/mL (or 37.5–75 nmol/L) was reported as vitamin D deficiency or insufficiency, with most of the studies indicating 20 ng/mL as the cut-off value [22,24,28,31,32].

### 3.4. Prevalence of MBD and Comparison with Control Population

Data on prevalence of MBD in children on long-term PN were largely heterogeneous, reflecting the different definitions used by the researchers. Prevalence of MBD (defined as BMD z-score ≤ −2) varied from 12.5%, reported by Ubesie [22] (*n* = 80), to 45% of IF children, reported by Poinsot [30] (*n* = 31). A total estimated prevalence of MBD of 28.8% (100/347 IF children) was calculated considering data from the nine studies using the cut-off of −2 z-score in at least one of the DXA parameter. A BMD z-score ≤ −1 was described in the vast majority of IF patients, with the highest prevalence reported by Diamanti [20] (83%, *n* = 24) and Olszewska [31] (87.5%, *n* = 17). BMD was found significantly reduced in IF children compared to controls in three studies [20,24,34], in a total of 55 IF children compared to 93 controls. This difference was not found by Dellert [19] in 1998, when BMC values (from 18 IF children and 36 controls) were adjusted for height and weight. Evidence of osteopenia based on standard radiography varied from 37.5% in the small study by Cannon [16] (*n* = 8), 59% in the study by Wozniak [28] (*n* = 27), and to 85.7% in the case series by Larchet [17] (*n* = 7).

Dependent on different cut-off and definitions, prevalence of vitamin D deficiency varied between 33% [29] and 63.8% [26] of IF children. Interestingly, no difference in serum levels of vitamin D were found in IF children compared to controls in one study [34], and a lower prevalence of vitamin D insufficiency in IF subjects was shown in another study [24].

Prevalence of fractures was described by few studies and varied from 3.8% in the study by Mutanen [23] (*n* = 41), 11.1% in the study by Demehri [26] (*n* = 36), 17% in the study by Diamanti [20] (*n* = 24), and 29% in the study by Khan [27] (*n* = 65).

### 3.5. Evolution of MBD and Risk Factors for MBD

Longitudinal studies based on repeated DXA studies reported data on 105 IF children (representing the 58% of the baseline cohorts of those studies) assessed after a period ranging from 1 to 6.2 years [20,24,26,29,30]. The three studies by Diamanti [20], Neelis [29], and Poinsot [30] (total *n* = 53) described an improvement of DXA parameters at follow-up. The study by Demehri [26] (*n* = 17) did not report significant changes in DXA status at a follow-up of 2 years, while the study by Pichler [24] reported a reduction of BMD at 1 and 2 years in 35 children with repeated DXA and a reduction of BMAD only at 1 year.

Several factors, including age, anthropometric data, the underlying gastrointestinal condition, dietetic factors (duration of PN, degree of dependency, calcium and vitamin D intakes), and bone markers level were found to be associated with MBD in the different studies.

Older age at DXA assessment was directly correlated with MBD in the studies by Ubesie [22] (*n* = 123) and Neelis [29] (*n* = 46) at a regression analysis. The longitudinal study of Poinsot [30] identified a younger age (at final DXA assessment, *n* = 31) as a factor associated with an increased prevalence of MBD at follow-up; however, effect of age was not analyzed at baseline. No correlation of BMD with age was found by Derepas [25] (*n* = 13) and Olszewska [31] (*n* = 17).

Increased weight and height for age were positively correlated to a higher BMD in the study by Diamanti [20] (*n* = 24) at a correlation analysis and in the study by Khan [27] (*n* = 65) for the weight z-score only at a multiple regression analysis. Patients with lower height z-score and higher weight for height (WFH) z-score had lower BMD z-scores in the study by Neelis [29] (*n* = 46), and this result was confirmed only for WFH z-score at a multivariate analysis. Lastly, no correlation with anthropometric measures was described by Olszewska [31] (*n* = 17) and by Derepas [25] (*n* = 13).

Some IF etiologies were correlated to a major risk of MBD, and this was reported for motility disorders and congenital enteropathy by Poinsot [30], for surgical IF (no SBS) by Neelis [29], and for non-SBS etiology by Diamanti [20]. Both the studies by Pichler [24] and Nader [33] showed a risk of lower BMD in children with congenital enteropathy, while only the study by Pichler [24] found higher risk in congenital motility disorders, but all these data were not confirmed at the logistic regression or multiple variate analysis. Duration of PN was found to be negatively correlated with BMD at multivariate analysis in the studies by Mutanen [23], Demehri [26], and Neelis [29] (total *n* = 123, PN duration, mean or median= 3.4–9.3 years). Dellert [19], Diamanti [20], Olszewska [31], and Nader [33] did not report a significant correlation between DXA parameters or risk of MBD and PN duration (total *n* = 99, PN duration, mean or median= 0.6–12.4 years). The study by Poinsot [30] did not find a correlation between PN duration and risk of MBD at baseline (*n*= 31, median PN duration= 2.7 years); however, at last DXA (*n* = 31, median PN duration = 9.2 years), children with MBD had shorter PN duration. The longitudinal analysis in the same study showed a reduced risk of MBD per year of PN (OR: 0.9 per year).

Dependence on PN at bone health evaluation was associated with reduced BMD in the study by Ubesie [22] at univariate analysis (data not confirmed at logistic regression analysis). The studies by Pichler [24] and Nader [33] did not confirm the dependency from PN or the entity (measures= number of PN days/week, PN dependency index) to be related to MBD. Duration of EN after weaning PN was a significant predictor for a lower BMD z-score in the population of 41 IF children, described by Mutanen [23]. Appleman [24] (*n* = 20) found no association between aluminum concentration in PN and reduced BMC or BMD.

A negative correlation between serum vitamin D levels and prevalence of MBD was found by Demehri [26] at univariate analysis but not confirmed at the multiple linear regression model. Levels of vitamin D or vitamin D deficiency were not related to the risk of MBD in the studies by Pichler [24] and Nader [33]. Lower calcium levels and higher PTH levels were more prevalent in patients with MBD in the study by Khan [27], but no influence of mineral status was found by Diamanti [20]. Calcium supplementation resulted as a predictor of MBD in the study of Mutanen [23].

## 4. Discussion

This is the first systematic review to define the prevalence and risk factors for development of MBD in children with IF. All the studies included in the review reported cases of MBD in IF children (based on radiologic techniques, mainly DXA), with percentages varying from 12.5% [22] to 87.5% [17].

The findings of this systematic analysis are largely heterogeneous. Reasons for heterogeneity include small and variable study sizes and control groups and large variability in the definition of intestinal failure and metabolic bone disease. Techniques of bone disease assessment were variable, and older studies were based on inaccurate and subjective methods (i.e., X-ray), while DXA is currently considered the gold standard for osteoporosis evaluation both in children and adults. Limiting the data to the studies based on DXA and in accordance with the current international pediatric definition [36], a “low bone mineral mass or density” (defined as BMC, BMD, or BMAD less than or equal to −2 z-score) was found in up to 45% of IF children [30]. As expected, reported prevalence increases when a less stringent cut-off (z-score ≤ −1) was adopted. Furthermore, considering the discrepancy of the results, data retrieved by this review clearly support the necessity of a bone assessment in the follow-up of children with IF. Prevalence data of MBD in IF children are similar to figures detected in adult populations. A large, multicenter study in adults with IF reported a prevalence of MBD between 31 and 41% according to the different definitions [10]. Prevalence up to 56% were reported by other studies [37,38]. Although these results support ours, a precise comparison is not possible considering the following: (1) The adult IF population differs largely from children for PN indications with a large proportion of IBD patients; (2) adult MBD has different definitions and different DXA site assessment; and (3) no systematic review on adult data has been conducted so far.

The studies included in the review pointed out several factors correlated to the risk of MBD, using different methodologies. Older children and subjects with growth failure and with some specific IF etiologies seem more at risk of developing MBD compared to other IF subgroups.

An inappropriate development of skeletal tissue or bone loss can be the consequence of an insufficient availability of micro and macronutrients in a malnourished patient; however, few data in this systematic review support the link between a poor nutritional status and a reduced BMD. The only two studies documenting this correlation did not introduce any correction of DXA measures for anthropometric parameters or for bone age, while several other studies corrected the DXA findings for measures of poor growth, retrieving more accurate figures of bone mass or density [22,23,24,29,30]. Some degree of overestimation of MBD in children with faltering growth cannot be excluded; therefore, current guidelines [36] recommend in cases of short stature or poor growth an adjustment for the height z-score or the calculation of the BMAD as a parameter that considers the bone size. Interestingly, in the two controlled studies [18,24], differences in BMC or BMD between cases and controls were no longer significant after correction for weight and height. Patients with IF are at high risk of growth failure during PN [35,39,40] and after PN cessation [19,29]; therefore, an adjustment of bone density parameters is strongly recommended.

Other factors have been advocated in the risk of reduced bone mass, such as the medical cause of IF and some PN-related factors, including nutrient and mineral deficiencies, excessive urinary calcium excretion, metabolic acidosis, and high aluminum concentrations in PN [41].

In our research, the underlying gastrointestinal condition leading to IF was found to be relevant by several studies. Specifically, patients with motility disorders (congenital intestinal pseudo-obstruction, long-segment Hirschsprung’s disease, and total or near-total aganglionosis) and congenital or early-onset enteropathies were at higher risk of developing MBD. The association with other conditions, such as nephropathy in some motility disorders, and use of steroids in some early-onset inflammatory enteropathies (autoimmune) can contribute to bone loss and ultimately to the severity of bone disease in these subgroups of IF patients, and therefore, a particular attention should be reserved to these subjects. Motility disorders and congenital enteropathies are also at risk for elevated fluid and electrolyte losses and consequent imbalance with the possibility of metabolic acidosis, another factor potentially impacting the bone mineralization process. In fact, a chronic acidotic state can directly impair vitamin D metabolism or compromise the bone buffering system while favoring the leaching out of calcium and phosphorus from the bone [35]. The impact of episodes of acidosis has been not specifically analyzed by the included studies.

Effects of mineral deficiencies on the BMD, mainly calcium and phosphorus, considered essentials for bone structure, were sporadically assessed in the studies included in the review. The negative correlation between the necessity of calcium supplementation [23] and a direct correlation between calcium serum levels and low BMD [27] identified by two studies suggest the importance of avoiding calcium deficiency and/or hypocalcemia. Furthermore, enteral calcium and phosphorus intakes were significantly lower in IF patients compared to healthy children [32], suggesting the necessity of supplementation. The right amount of parenteral supplementation of calcium and phosphorus has long been debated considering the risk of calcium-phosphate precipitation in PN solution, and no specific guidelines on enteral calcium and phosphorus intakes are available.

The relationship between vitamin D and PN-associated MBD is less evident. If vitamin D deficiency is clearly associated with risk of bone loss and rickets, patients with IF have several reasons to be at risk of hypovitaminosis D (malabsorption, reduced sun exposure, etc.); conversely, the risk of vitamin D toxicity has been raised in patients on long-term PN. The excess of vitamin D would increase bone reabsorption in these patients and consequently contribute to bone disease. Several studies in adults on PN demonstrate an improvement in BMD after vitamin D reduction or withdrawal without consequences on vitamin D serum levels or calcium balance [41,42]. One study in IF children showed no effect of discontinuing vitamin D in the serum levels of 1,25-(OH2)D and calcium/phosphorus balance, with no clinical consequences [17]. According to the results of our review, vitamin D deficiency was common in IF patients; however, no significant differences were seen with the control population, and no clear evidence of a relation between vitamin D levels and risk of MBD emerged. Overall, vitamin D requirement does not seem to be increased in patients with IF (unless other comorbidities are present, such as renal or liver-cholestatic disease), and an adjustment based on serum levels is recommended.

The role of specific PN mixtures or components in the risk of MBD was not systematically investigated in the included studies. The contribution of aluminum toxicity on bone metabolism and MBD development has been advocated for several years [35]. In the last years, low-containing PN solutions have been developed, reducing such risk, which remains relevant in patients with concomitant kidney impairment. Only one study [24] evaluated this association, showing, despite an elevated serum aluminum concentration in IF subjects, a lack of correlation with BMC or BMD.

Overall results from the studies included in the review suggest the risk of MBD in children both during and after PN cessation. Unexpectedly, controversial results were found on the association between degree of PN dependence and/or duration of PN with the severity of bone disease, not permitting to draw firm conclusion on this argument. This reinforces the multifactorial pathogenesis of MBD in intestinal failure. Despite the evidence of bone disease in IF children several years after withdrawing PN, the reduction of the risk of MBD in the longitudinal study by Poinsot [30] is encouraging. However, overall results of our review emphasize the importance of a regular and long-term monitoring. The most recent guidelines on pediatric HPN suggest measurement of BMD using DXA on a 2–3 yearly basis or annually if previously abnormal [43]. Strategy of primary and secondary prevention are fundamental also considering that treatment options are restricted and sporadically reported in IF children [44,45].

Some limitations to this systematic review need to be mentioned. First, the quality of the studies varied considerably; most of the studies were small and retrospective, and the statistical analysis not always included a multivariate or regression analysis. Studies were heterogeneous in patient selection (definition of the IF population) and outcome measures, particularly in the definition of bone disease and vitamin D deficiency. For these reasons, the possibility to perform a metanalysis was not considered. Studies were limited to European and U.S. centers; therefore, data could be not representative of other countries. All these limitations highlight the need for prospective, multicenter studies on this topic based on standardised DXA measures, corrected for growth failure, and compared with population nomograms.

## 5. Conclusions

In conclusion, data from our review indicate an increased prevalence of metabolic bone disease in children with intestinal failure. The risk is higher in older children with growth failure and longer PN duration; however, the possibility of MBD remains significant even when the enteral autonomy is achieved. A long-term monitoring of bone health through DXA assessment (with adequate correction for anthropometric measures) and a careful evaluation of calcium balance and vitamin D requirement is warranted in children with IF. Given the importance of the topic and the limitations of the published studies, there is a need for prospective population multicenter research on this argument. Specifically, the impact of single PN components and the effects of treatment strategies should be assessed in a longitudinal, prospective pediatric studies.

## Figures and Tables

**Figure 1 nutrients-14-00995-f001:**
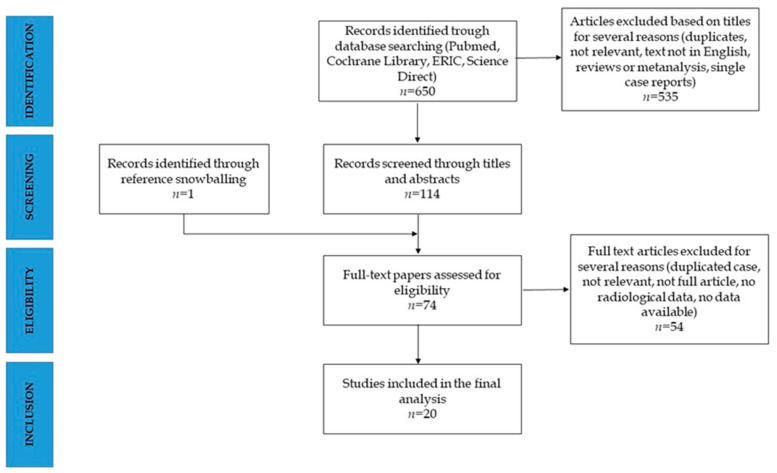
PRISMA diagram for the systematic search.

**Table 1 nutrients-14-00995-t001:** Quality assessment results, based on Quality Criteria Checklist: Primary Research.

	Cannon RA, 1980 [16]	Larchet M, 1991 [17]	Leonberg BL, 1998 [18]	Dellert SF, 1998 [19]	Diamanti A, 2010 [20]	Olieman JF, 2012 [21]	Ubesie AC, 2013 [22]	Mutanen A, 2013 [23]	Appleman SS, 2013 [24]	Pichler J, 2014 [35]	Derepas C, 2015 [26]	Demehri FR, 2015 [27]	Khan FA, 2015 [28]	Wozniak LJ, 2015 [29]	Neelis E, 2018 [30]	Poinsot P, 2018 [31]	Olszweska K, 2018 [32]	Kvammen JA, 2020 [33]	Nader EA, 2021 [34]	Louazon T, 2021 [36]
Relevance questions																				
1. Would implementing the studied intervention or procedure (if found successful) result in improved outcomes for the patients/clients/population group? (NA for some Epi studies)	N/A	N/A	N/A	N/A	Yes	Yes	Yes	Yes	Yes	Yes	Yes	Yes	Yes	Yes	Yes	Yes	Yes	Yes	Yes	Yes
2. Did the authors study an outcome (dependent variable) or topic that the patients/clients/population group would care about?	Yes	Yes	Yes	Yes	Yes	Yes	Yes	Yes	Yes	Yes	Yes	Yes	Yes	Yes	Yes	Yes	Yes	Yes	Yes	Yes
3. Is the focus of the intervention or procedure (independent variable) or topic of study a common issue of concern to dietetics practice?	Yes	Yes	Yes	Yes	Yes	Yes	Yes	Yes	Yes	Yes	Yes	Yes	Yes	Yes	Yes	Yes	Yes	Yes	Yes	Yes
4. Is the intervention or procedure feasible? (NA for some epidemiological studies)	Yes	Yes	Yes	Yes	Yes	Yes	Yes	Yes	Yes	Yes	Yes	Yes	Yes	Yes	Yes	Yes	Yes	Yes	Yes	Yes
Validity questions																				
1. Was the research question clearly stated?	Yes	Yes	Yes	Yes	Yes	Yes	Yes	Yes	Yes	Yes	Yes	Yes	Yes	Yes	Yes	Yes	Yes	Yes	Yes	Yes
2. Was the selection of study subjects/patients free from bias?	No	No	No	Yes	Yes	Yes	Yes	Yes	Yes	Yes	Yes	Yes	Yes	Yes	Yes	Yes	Yes	Yes	Yes	Yes
3. Were study groups comparable?	N/A	N/A	N/A	Yes	Yes	N/A	N/A	Yes	Yes	Yes	Yes	N/A	N/A	Yes	Yes	Yes	N/A	Yes	Yes	Yes
4. Was method of handling withdrawals described?	Yes	Yes	N/A	N/A	Yes	Yes	Yes	Yes	N/A	Yes	N/A	Yes	No	Yes	Yes	Yes	Yes	N/A	Yes	Yes
5. Was blinding used to prevent introduction of bias?	No	No	No	Unclear	No	No	no	No	Yes	No	Yes	No	No	Yes	No	No	no	no	no	no
6. Were intervention/therapeutic regimens/exposure factor or procedure and any comparison(s) described in detail? Were intervening factors described?	Yes	Yes	Yes	Yes	Yes	Yes	Yes	Yes	Yes	Yes	Yes	Yes	Yes	Yes	Yes	Yes	Yes	Yes	Yes	Yes
7. Were outcomes clearly defined and the measurements valid and reliable?	Yes	Yes	Yes	Yes	Yes	Yes	Yes	Yes	Yes	Yes	Yes	Yes	Yes	Yes	Yes	Yes	Yes	Yes	Yes	Yes
8. Was the statistical analysis appropriate for the study design and type of outcome indicators?	Yes	Yes	Yes	Yes	Yes	Yes	Yes	Yes	Yes	Yes	Yes	Yes	Yes	Yes	Yes	Yes	Yes	Yes	Yes	Yes
9. Are conclusions supported by results with biases and limitations taken into consideration?	Yes	Yes	Yes	Yes	Yes	Yes	Yes	Yes	Yes	Yes	Yes	Yes	Yes	Yes	Yes	Yes	Yes	Yes	Yes	Yes
10. Is bias due to study’s funding or sponsorship unlikely?	Yes	Yes	Yes	Yes	Yes	Yes	Yes	Yes	Yes	Yes	Yes	Yes	Yes	Yes	Yes	Yes	Yes	Yes	Yes	Yes
TOTAL SCORE	Neutral	Neutral	Neutral	Positive	Positive	Neutral	Neutral	Positive	Positive	Positive	Positive	Neutral	Neutral	Positive	Positive	Positive	Neutral	Positive	Positive	Positive

**Table 2 nutrients-14-00995-t002:** Main characteristics of the included studies.

First Author, Year	Study Details (Study Type, Participants, Years, Country)	Intestinal Failure/pn- Dependence Definitions	Bone Disease Measurement and Definitions	Study Population (Number, Age, Sex, Length of pn)	Control Group or Reference Population	Prevalence of Bone Disease and/or Comparison with Control Group	Vitamin d Status	Follow-Up Results	Factors Associated with Bone Density or Metabolic Bone Disease
Cannon RA, 1980 [16]	Observational, single-center study (OSCS). Children on HTPN at August 1979 with a follow-up of 2 years. USA.	TPN begun within 2 months of age and still ongoing.	Development of pathologic fractures or loose teeth plus radiologic evidence of rickets were assessed.	8 children, age 7–24 months, 5 males. Duration of TPN: 7–24 months.	No	37.5% had bone disease.	Supplementation with vitamin D resulted in healing of bone lesions in one patient.	Not evaluated	Not reported
Larchet M, 1991 [17]	OSCS. Children on cyclic HPN at November 1983, with a follow-up of 4 years. France.	HPN for SBS.	Osteopenia assessed on wrist and tibia X-ray. Sserum Ca, P, ALP, PTH, and urinary Ca and P were annually measured. Bone biopsy.	7 children, age 4–14 years, 4 males. Duration of PN at study start: 19–79 months.	No	85.7% had signs of moderate osteopenia. The cortico-diaphyseal indices of the tibia were normal (>0.5).	Mean 25-(OH)D levels dropped from 23 ± 6 ng/mL (1983) to 5.5 ± 2 (1987), (*p* < 0.03).	No change on radiologic signs of osteopenia during the 4 years of the study.	Not reported
Leonberg BL, 1998 [18]	OSCS. Children with IF, weaned off PN in the previous 7 years (at least since 6 months) evaluated for a nutritional and bone disease screening. USA.	PN for more than 4 months, beginning in the 1st postnatal week, including 1 month of HPN.	BMC of the radius measured by single-photon absorptiometry, compared with age appropriate norms.	9 children, age 2.8–6 years (mean, SD: 4.9 ± 1), 5 males. Duration of PN: 5–39 months (mean, SD: 14.6 ± 11.4). Duration of HPN: 8.5 ± 5 months. Mean time of PN cessation: 3.4 ±1.4 years	18 healthy controls matched for age, gender, and race.	Total BMC of SBS children was reduced compared to controls but not different after correction for weight and height (*p* = 0.8). 44.4% had BMC below normal.	Median 25-(OH)D levels: 26 ng/mL.	Not evaluated	75% subjects with low BMC were consuming diets deficient in calcium. No correlation of BMC with PN duration or time since discontinuation.
Dellert SF, 1998 [19]	OSCS, case-control. Children with SBS, totally weaned from PN and with a bone disease screening (1984–1994). USA.	SBS (intestinal resection, omphalocele, gastroschisis) receiving at least 1 month of PN.	BMC measured by DXA in subjects and controls. Serum Ca and P levels were measured.	18 children, age 2–8 years, 9 males. Duration of PN: 1–67 months (median 7 months). Discontinuation of PN: 1–8 years (median 3 years).	36 healthy controls matched for age and gender.	BMC of cases was reduced compared to controls. After adjustment for weight and height, there was no difference in BMC between SBS and controls. Serum Ca e P were comparable.	Mean serum 25-(OH)D were lower in children with SBS respect to controls (26.0 ± 9.8 ng/mL vs. 41.3 ± 8.4 ng/mL, *p* = 0.002); mean serum 1,25-(OH)_2_D were similar.	Not evaluated	BMC was not correlated with PN duration or time since PN discontinuation.
Diamanti A, 2010 [20]	OSCS, case-control. Children with IF on a BMD evaluation program (September 2005-September 2007). Italy.	IF: condition needing PN providing at least 75% of total calories for >4 weeks or at least 50% calories for >3 months.	BMD (BMC, areal BMD, BMAD, BMD z-score) measured by DXA at baseline in patients and controls and after 1 year (in 9 patients). BMC reduction: BMD z-score <−1. Development of fractures was assessed.	24 children, age 3.5–17.5 years (6.7 ± 5.2 years), 14 males. 7 off PN. Duration of HPN: 3–181 months (median 38).	24 healthy controls matched for age and gender. Mean age: 6.5 ± 3.9 years, 18 males.	83% had BMD z-score ≤ −1. BMD z-score, BMD and BMAD were significantly lower in patients compared to controls (*p* < 0.05 and *p* < 0.01). Subjects with BMD z-score >−1 were lower in patients (17%) compared to controls (50%). 17% developed fractures.	Not evaluated.	All DXA variables were increased at the 2nd DXA in 7/9 patients at 1 year follow-up.	No correlation between bone mineral status and PN duration and nutrient intake. Significant correlation between BMC and BMD and weight and height. BMD z-scores were higher in SBS-IF compared to medical causes of IF.
Olieman JF, 2012 [21]	OSCS, cross-sectional. Children with infantile SBS (within 1 year) and off PN invited for a nutritional/bone disease screening (January 1975–January 2003). Netherland.	IF: 70% of resection of SB or PN needed for >42 days after resection or residual SB length <50 cm for preterm and <75 cm for term neonates	BMD LS, BMD TB, BMC, LBM, and %BF were measured by DXA and compared to Dutch reference data.	40 subjects, 16 males, mean age 14.8 ± 6.8 years, including 31 children, mean age 11.8 ± 4.2 years. Duration of PN: median 110 days (43–2345)	Dutch reference population.	In children, mean BMC, LBM*, and BMD LS were lower than the reference values (*p* < 0.05).	Not evaluated.	Not evaluated	Not reported
Ubesie AC, 2013 [22]	OSCS, retrospective. IF patients aged ≥3 years under nutritional follow-up in a 5-year period (2007–2012). USA.	IF: need for PN support for more than 30 days.	Lumbar spine BMD (L1–L4) was measured by DXA (within 6 months from serum evaluation) and compared to reference values. BMD z-score were adjusted for age and height for “short” patients (height <5th percentile on CDC charts). Reduced BMD: BMD ≤ −2 z-score.	123 IF patients, median age: 4 years (3–22 years), 71 (57.7) % males. PN duration not indicated. 80 had a DEXA scan.	Reference population (not specified).	12.5% had a low BMD z-score (adjusted for height).	40% had vitamin D deficiency (25-(OH)D < 20 ng/mL).	Not evaluated	PN dependence at bone health evaluation was associated with reduced BMD. Increased age was significantly associated with vitamin D deficiency (*p* = 0.04) and reduced BMD z-score (*p* = 0.01) at a binary logistic regression analysis. At a logistic regression model, age >10 years was associated only to reduced BMD (*p* = 0.02).
Mutanen A, 2013 [23]	OSCS, cross sectional. Children with IF invited to a bone disease screening (January 1984–August 2010). Finland.	IF: 50% resection of SB or duration of PN >30 days.	BMD in the left proximal femur and lumbar spine was measured by DXA, and compared with reference values. BMD z-score ≤ −1 was considered low. BMD z-scores were corrected for bone age. Vertebral compression >20% was considered abnormal. PTH levels were measured. Occurrence of fractures was assessed.	41 IF children, 27 males, mean age 9.9 years (0.2–27). PN duration 41 months (0.7–250). 30 off PN, time after weaning off: 9 years (0.3–27). 30 had a DXA study.	Reference population (not specified).	BMD z-score was ≤−1 in 70% and ≤−2 in 43%. 3.8% had sustained fractures. 3.8% had vertebral compression.	41% had vitamin D deficiency (25-(OH)D < 15 ng/mL), and 44% had secondary hyperparathyroidism. Mean 25-(OH)D levels: 21.6 ng/mL on PN; 23.6 ng/mL in patients off PN (*p* = 0.642).	Not evaluated	At the multiple regression analysis duration of EN after weaning PN, duration of PN and calcium supplementation were the significant predictors for a lower lumbar spine BMD z-score.
Appleman SS, 2013 [24]	Observational, case-control, double-center study. PN-dependent IF children invited to a bone disease screening. USA.	PN-dependent IF: inability to sustain growth without PN	BMC and BMD of the lumbar spine were measured by DXA. PTH was measured.	20 IF children, median age 26 months (6–127), 9 males (45%). Median time on PN: 18.5 months (4–103)	49 healthy controls median age 25 months (7–127), 22 males (45%).	Lumbar spine BMC was 15% lower (*p* = 0.0009), and BMD was 12% lower (*p* = 0.0035) in IF subjects compared to controls. No differences after adjusting for weight and height.	IF participants had higher serum 25(OH)D than controls (mean levels: 40 vs. 30 ng/mL, *p* = 0.0005). IF patients had lower prevalence (5% vs. 14%) of vitamin D insufficiency (25(OH)D < 20 ng/mL).	Not evaluated	No association between aluminum concentration and BMC or BMD.
Pichler J, 2014 [35]	OSCS. Children aged >5 years with IF and being on HPN (2002–2010). UK.	IF: HPN for at least 6 months.	BMD and BMAD were calculated by DXA + bone age assessed by X-ray. Low BMD or BMAD: ≤−2 SDS. Biochemical markers (also ALP) were measured. Occurrence of fractures was recorded.	45 subjects, 24 males, age 7.7 years (5–18). Time of HPN: 5 years (3.2–12). 18 (40%) on TPN, 26 (58%) on PPN, 1 (2%) off PN.	UK reference data.	42% had BMD -≤−2 SDS and 31% had BMAD ≤−2 SDS. Mean BMD SDS was −1.7 ± 1.6; mean BMAD SDS was −1.4 ± 1.5. 37% had a history of fractures (median 1.3), all non-pathological.	Mean 25(OH)D levels: 57.6 ± 6.44 ng/mL. 7% patients had vitamin D insufficiency (25(OH)D < 25 ng/mL).	35/42 had repeated DXA at 1 and 2 years. BMD declined significantly at 1 and 2 years, while BMAD only at 1 year. In a multivariate model, the only significant predictor of a change in the BMD SDS was a change in the weight SDS.	No difference of BMD and BMAD SDS according to diagnosis, degree of dependence on PN, and presence of 25-(OH)D deficiency. In a multivariate model, age at the time of DXA was the only predictor of BMD SDS.
Derepas C, 2015 [26]	OSCS, case-control. IF children on HPN (June 2012–August 2012). Canada.	IF: need of PN providing at least 25% calories for more than 6 weeks.	BMD measured annually by DXA on the lumbar spine (L1–L4) and compared to reference values. Reduced BMD: BMD z-score ≤ −1. Serum OC, CTx, BSAP, and PTH levels were assessed.	13 IF children, age 1.2–10.7 years, 12 males. Duration of PN: 0.5–12.5 years. 9 subjects had DXA (7 results available)	20 healthy controls matched for age and gender.	Mean serum OC and CTx concentration were lower in IF compared to controls (*p* < 0.01 and *p* < 0.05). No differences in BSAP and PTH levels. BMD measured in 9/13 IF subjects ranged from 0.401 to 0.838 g/cm^2^. BMD z-score ranged from −3.526 to +1.5. 57% subjects had BMD z-score ≤ −1. 28.5% had BMD z-score ≤ −2.	Not evaluated	Not evaluated	A significant inverse relation was found between BMD and serum OC but no with length of PN, age, weight, or height z-score. BMD z scores was inversely related to CTx also after controling for weight z-scores.
Demehri FR, 2015 [27]	OSCS, retrospective. Patients with IF (2006–2012) who underwent DXA screening (at 6 years). USA	IF: PN dependence of at least 60 days.	BMD measured by DXA at the lumbar spine (L1–L4). BMD z-scores were derived from reference values. MBD was defined as BMD z-score ≤ −1. Serum Ca, P, and PTH measured within 4 weeks of DXA scan. History of pathologic fractures and bone pain.	36 IF subjects, 21 males 25 off PN (69.4%). Mean PN duration 5.1 ± 5.4 years. 17 subjects had a repeated DXA.	Reference U.S. population.	Mean BMD z-score was −1.16 ± 1.32. 50% had MBD. 11.1% had pathologic fractures, and 16.6% had history of bone pain.	Mean 25(OH)D levels: at 1st DXA, 25.11 ± 13.05 ng/mL; at 2nd DXA, 23.67 ± 12.73 ng/mL (*p* = 0.821). 63.8% had vitamin D deficiency (25(OH)D ≤30 ng/mL), and 25% had hyperparathyroidism (>55 pg/mL). No change in serum 25-(OH)D (*p* = 0.821), despite an increase in % of subjects on vitamin D supplementation.	In 17/36 with repeated DXA, no change at the 2nd DXA (after 2 ± 1.1 years) occurred (*p* = 0.199).	Duration of PN and serum 25-(OH)D were predictors of BMD z-score at univariate analysis. In a multivariate analysis, the only significant predictor of a reduced BMD z-score was the length of PN (B = −0.132, *p* = 0.006).
Khan FA, 2015 [28]	OSCS, retrospective. Patients with IF who underwent DXA screening at 5 years of age (2004–2013). USA.	Not reported.	Whole-body DXA was performed, and BMD z-scores were determined using normative data. For patients with repeated DXA scans, the lowest BMD z-scores were recorded. MBD: BMD z-score ≤ −2. Serum levels Ca, P, and PTH were measured. History of fractures was recorded.	65 IF subjects, 34 males. 39 off PN (60%). Mean PN duration 44.2 ± 43.2 months.	Reference U.S. population.	BMD z-score ≤ −2 was reported in 34%. 29% experienced at least 1 fracture.	Mean 25(OH)D levels: 27 ± 42 ng/mL. 42% had vitamin D deficiency (25(OH)D ≤30 ng/mL).	Not evaluated.	At univariate analysis, patients with BMD z-score ≤ −2 had lower WAZ (*p* = 0.01), lower Ca (*p* = 0.04), and higher PTH levels (*p* = 0.006). At the multivariable logistic regression analysis, WAZ (R = 1.8, *p* = 0.03) and serum Ca (R = 3.8, *p* = 0.02) were independent factors of a low BMD z-score.
Wozniak LJ, 2015 [29]	OSCS, retrospective. IF children under follow-up (January 2012–June 2012) who underwent at least 1 determination of serum 25-(OH)D between July 2010 and June 2012. USA.	IF: HPN required for at least 6 months.	Osteopenia evaluated on subjective analysis of standard radiology studies (bone mineralization of axial skeleton, long bones, and blurring of the cortical white line). History of fractures was assessed.	27 IF children, median age 5.5 (IQR: 2.7–8.2 years), 12 (44%) males. Median PN duration: 3.5 years (IQR: 2.6–6.9).	No	59% had osteopenia. 11% had fractures.	41% had vitamin D insufficiency (25(OH)D = 20–29 ng/mL); 1 child had vitamin D deficiency (25(OH)D < 20 ng/mL). Supplementation with vitamin D resulted in improvement of 25-(OH)D levels (2/27) and in improvement at blood drawn following the study period (4/27).	Not evaluated.	At univariate and multivariate logistic regression analysis, the only variable associated with 25-(OH)D levels was diagnosis of SBS. Duration of PN was not correlated.
Neelis E, 2018 [30]	OSCS, retrospective. IF children (followed between 2000 and 2015) who underwent at least one DXA or DXR. The Netherlands.	IF: PN needed for >6 weeks.	Total-body and lumbar spine (L2, L2–4) BMD was measured by DXA after 4–5 years of age. BMD z-scores were determined by national reference values. BMD was adjusted to the bone size calculating the BMAD. Low BMD or BMAD: z-score ≤ −2. DXR was used to calculate the BHI that was compared to a reference population. History of fractures was recorded.	46 IF subjects, 20 (44%) males. Age at 1st DXA: 6 years (IQR: 5.5–9.9). 28 off PN (76%). Median PN duration: 9.4 months (IQR: 4.6–14.3). 37 had 1st DXA. 13 children had multiple DXA scans after a median time of 2.1 years (1.09–2.44).	Reference Dutch population.	24.3% had a low BMD (either BMD TB, LS or BMAD z-score ≤ −2) at 1st DXA. Median BMD TB, BMD LS, and BMAD z-scores were significantly lower compared to the reference population (*p* = 0.006; *p* < 0.001 and *p* = 0.004). 50% had low BHI (z score ≤ −2) at 1st radiography. 4/46 children developed multiple fractures (100% were PN dependent at time of 1st fracture).	33% had vitamin D insufficiency (25(OH)D < 20 ng/mL); no differences between children on PN or weaned off.	Median change in z-scores per year was +0.16 SD for BMD TB and +0.09 for BMD LS.	Patients still on PN at 1st DXA had lower median BMD TB z-score (*p* = 0.048), and the proportion of children with BMD TB z-scores ≤ −2 was higher (*p* = 0.008). At univariate analysis, an older age, a lower HFA z-score, a HFA z-score ≤ −2, a higher WFH z-score, and a longer PN duration were related to lower BMD LS z-scores. In a multivariate model older age, longer duration of PN and surgical IF were related to lower z-scores.
Poinsot P, 2018 [31]	OSCS, retrospective. IF children (January 2004–January 2014) on HPN with at least 2 DXA (on HPN since at least 2 years from last DXA). France.	IF: children on HPN for at least 6 months.	TBMC, LTM, and FM were measured by DXA and adjusted for ideal WFH. LBM** was defined as a TBMC z-score ≤ −2, following the reference values. Serum Ca, P, and ALP were measured within 3 months of DXA assessment.	31 children, 14 males (45%). Median age at 1st DXA: median age: 2.9 years (0.4–13.3). Median PN duration: 2.7 years (0.1–12.7). Median age at last DXA 11.4 years (3.7–19.7), and median PN duration: 9.2 years (2.9–19.6). Median time between 1st and last DXA: 6.2 years (0.7–16.6).	Reference population composed of 68 healthy children (2–24.9 years, 31 males) and 55 newborns (gestational age 33–40 weeks).	45% had LBM** at 1st DXA. Median TBMC z-score at 1st DXA was −1.9 SD (−5.3–2.6). 71% had TBMC z-score for ideal WFH ≤−1.	Median 25(OH)D levels: at 1st DXA, 15.4 ng/mL; at last DXA, 14.0 ng/mL.	TBMC z-score adjusted for ideal WFH increased significantly at last DXA (+0.1 ± 0.04 per year, *p* = 0.012). At last DXA, 29% had LBM (*p* = 0.197 with 1st DXA).	IF etiology (CIPO, HD, and CE) and a lower plasmatic creatinine level were related to LBM** prevalence at baseline. At last DXA, LBM** was associated with a shorter PN duration (*p* = 0.045) and a younger age (*p* = 0.023). The risk of the LBM** decreased significantly with the duration of HPN (OR 0.9 per year of PN, *p* = 0.018).
Olszewska K, 2018 [32]	OSCS. Patients with ultra-short bowel syndrome on HPN under nutritional follow-up. Poland.	IF: residual small bowel <10 cm at the time of resection (first 2 months of life) and receiving PN for at least 6 months.	Antero-posterior spinal and total bone mass density measured by DXA, and BMD expressed as z- scores. Decreased BMD TB: ≤−1 z-score. Abnormal bone mineralization: BMDs ≤−1 z-score.	17 children, age 0.8–14.2 years (median 6.6), 9 males on HPN. Median PN duration: 6.6 years (0.8–14.2 years). 8 subjects had DXA.	Reference population (not specified).	50% had BMD ts ≤ −1 z-score, and 87.5% had BMD s ≤ −1 z-score.	Mean 25(OH)D levels: 20.1 ng/mL. 53% had vitamin D deficiency (25(OH)D < 20 ng/mL).	Not evaluated.	BMD z-scores were not correlated with body mass or body height SDS, age or PN length.
Kvammen JA, 2020 [33]	OSCS, cross sectional, controlled. Children with IF on HPN (March 2017–September 2017) who underwent a bone disease screening. Norway.	IF: children dependent on HPN for >6 months.	BMD was measured by DXA total body (TB) and spinal (LS at L2–L4). BMD z-scores were calculated according to reference values and corrected for bone age (if height was <−2 z-scores). Suboptimal BMD: BMD ≤−1 to −1.99 z-score. Low BMD: ≤−2 z-score.	19 IF children, mean age: 10.1 years, SD: 3.5, 13 males (68%). 100% on HPN, 4 on TPN. 12 IF children had DXA.	Reference population composed of 50 healthy controls, mean age: 10 years, SD: 3.6, 18 males (36%). 46 controls had DXA.	IF group had significantly lower median BMD z-score for total body (*p* < 0.001) and lumbar spine (*p* = 0.01). 25% IF subjects had suboptimal BMD TB, 17% had suboptimal BMD LS (vs. 5% of healthy controls), and 25% of IF children had low BMD.	IF group had lower 1,25-(OH)_2_D levels compared to controls (29.6 vs. 57.5 pg/mL, *p* < 0.001). IF patients and healthy had similar level of 25(OH)D (28.4 vs. 32.4 ng/mL, *p* = 0.29).	Not evaluated	Not reported
Nader EA, 2021 [34]	OSCS, retrospective. IF children >5 years on HPN, who underwent a DXA screening (January 2016–December 2018). France.	IF: HPN for at least 2 years	BMD of the lumbar spine (L1–L4), left femur, and total body were measured using DXA and expressed as z-scores (compared to reference values). BMD values were adjusted to height or statural age. Low BMD: ≤−2 z-score.	40 IF children, 24 males, median age 12.4 ± 4.5 years. HPN duration 12.4 years ± 4.4.	Reference population (not specified).	BMD LS ≤−2 z-score: 30% BMD LS adjusted ≤−2 z-score: 18% LF BMD ≤−2 z-score: 15% LF BMD adjusted ≤−2 z-score: 15% BMD TB ≤−2 z-score: 23% BMD TB adjusted ≤−2 z-score: 18%	Mean 25(OH)D levels: 26.5 ± 9.1 ng/mL; Median 25(OH)D levels: 25 ng/mL.	Not evaluated	No correlation between indication of PN, duration, and degree of PN dependency and level of 25-(OH)D3.
Louazon T, 2021 [36]	Cross-sectional, case-control, single-center study. IF children, aged >9 years, on HPN who underwent a bone assessment (March 2014–June 2015). France	IF: HPN for at least 2 years	vBMD was evaluated by HR-pQCT at the non dominant limb. BMC and BMD were measured by TB and spine (L1–L4) DXA. Serum NMID osteocalcin and PTH were measured.	11 IF children, 8 males, median age: 16 years (9–19). Median PN duration: 10.3 years (6.4–18.3). 2 off PN.	20 healthy controls, 16 males, median age: 16 years (10–17).	In IF children, increased PTH (*p* = 0.003) and reduced OC-diasorin (*p* = 0.005). At the tibia, IF children had low trabecular area compared to controls (*p* = 0.003). BMD TB was lower in IF subjects (*p* = 0.039). 18% of IF patients had BMC TB ≤−2 DS.	No differences in mean 25-(OH)D3 levels between patients and controls (17.6 vs. 22.8 ng/mL, *p* = NS).	Not evaluated	Not reported

(H)TPN, (home) total parenteral nutrition; (H)PN, (home) parenteral nutrition; IF, intestinal failure; Ca, calcium; P, phosphorus; ALP, alkaline phosphatase; PTH, parathyroid hormone; 25-(OH)D, 25-hydroxyvitamin D; (T)BMC, (total) bone mineral content; SB, short bowel; SBS, short bowel syndrome; BMD, bone mineral density; BMAD, bone mineral apparent density; DXA, dual-energy X-ray absorptiometry; BMD LS, bone mineral density of the lumbar spine; LBM*, lean body mass; BMD SDS, age and sex-adjusted bone mineral density; LTM, lean tissue mass; EN, enteral nutrition; CDC, Center for Disease Control; OC, osteocalcin; CTx, c-telopeptide; BSAP, bone-specific alkaline phosphatase; WAZ, weight-for-age z-score; MBD, metabolic bone disease; BMD TB, bone mineral density total body; HFA, height for age; BHI, Bone Health Index; WFH, weight for height; LBM**, low bone mass; LF BMD, left femur bone mineral density; HR-Pqct, high-resolution peripheral quantitative computed tomography; NMNID, N-terminal mid fragment.

## Data Availability

The data presented in this study are available on request from the corresponding author.

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
