# Peer review of "Metabolic Bone Disease in Children with Intestinal Failure and Long-Term Parenteral Nutrition: A Systematic Review"

_nutrients, 2022, doi:10.3390/nu14050995_

Round 1
Reviewer 1 Report
The authors performed a systematic review on MBD in children with IF, defined as being dependent on PN for some time. The subject is important, and as they mention, this is the first systematic review.
The introduction is overall clear, except when it comes to defining MBD. Please rewrite line 52-55: what is the difference between abnormal calcification and decrease in bone mineralization? Please explain the differences between BMC, BMD and BMAD in the introduction more clearly (I did not find the full wording of BMAD in the text). line 70 is confusing: how can metabolic markers "drive a possibility"? Please reformulate. Line 72 please add: "While in adults dependent on parenteral nutrition..."
Methods are clearly described. Please explain why you choose to include such a long period, taking into account major changes in parenteral nutrition regimes between 1980 and 1995-2000. Why not limit to last 25 years? results from 1980 are most probably no longer relevant. (minor textual remark, line 88: "results were as (the) follows").
Table 2 contains a lot of results and is difficult to read. I propose to turn the table 90° (landscape format) and to make a separate column for presentation of vit D results (is it possible to give median and range of levels of 25 OH vit D and proportion under 20 ng/ml for each study?). Results can also presented in a more uniform way for the different studies, making it more easy for the reader to compare (eg instead of "vit D was measured" for one study, and "insufficiency 25 OH vit D level < 30 ng/ml" for another one, please indicate whether 25 OH vit D measurement was included and which were the results // Same for method for evaluation of bone health, which results).
line 209: "Dependent on according to"
The discussion needs more structure. Try to summarize the results (what is known, what remains unclear) and confront them with results in adults (as is done already in the actual manuscript). How to explain the heterogeneity of the results. Please highlight the heterogeneity of definitions and of the study methods, the lack of information on the content of PN mixtures and protocols used for administration of TPN, and try to end with conclusions for further primary research. How to standardize definitions and methods for the evaluation of bone health?
Line 292-299 is not clear to me. As no information is included on the content of PN mixtures, the discussion in line 333-348 is not really linked to the results presented. The same holds true when mentioning biphosphonates and Denosumab (line 376-383). Try to limit the discussion to what is presented in the result section, or clearly explain why these items should be included in future research on the topic.
Try to be more specific in the conclusion regarding the kind of research needed.
Author Response
We thank the reviewers for their careful and meaningful evaluation of our manuscript, and for their important comments. The manuscript has been revised according to the suggestions of the referees. Please find our point-by-point reply to referees’ comments below.
Reviewer 1
The authors performed a systematic review on MBD in children with IF, defined as being dependent on PN for some time. The subject is important, and as they mention, this is the first systematic review.
The introduction is overall clear, except when it comes to defining MBD. Please rewrite line 52-55: what is the difference between abnormal calcification and decrease in bone mineralization? Please explain the differences between BMC, BMD and BMAD in the introduction more clearly (I did not find the full wording of BMAD in the text). line 70 is confusing: how can metabolic markers "drive a possibility"? Please reformulate. Line 72 please add: "While in adults dependent on parenteral nutrition..."
Thanks for your suggestions. We have now clarified the spectrum of MBD (lines 51-62) and explained more clearly DXA measures in the introduction session (lines 73-77). We have changed lines 70 and 72 according to your comments (lines 80-82).
Methods are clearly described. Please explain why you choose to include such a long period, taking into account major changes in parenteral nutrition regimes between 1980 and 1995-2000. Why not limit to last 25 years? results from 1980 are most probably no longer relevant.
In order to evaluate whether progress in PN composition and IF management would have impacted the development of complications (such as MBD), we decided to include a long period of observation. Actually the original research retrieved about 50% of articles published before’ 2000 and 4 were included in the final analysis. Considered the paucity of data on this argument we felt that the inclusion of 4 studies (total of 42 subjects) would have been an added value.
(minor textual remark, line 88: "results were as (the) follows").
Thanks, we have changed the sentence.
Table 2 contains a lot of results and is difficult to read. I propose to turn the table 90° (landscape format) and to make a separate column for presentation of vit D results (is it possible to give median and range of levels of 25 OH vit D and proportion under 20 ng/ml for each study?). Results can also presented in a more uniform way for the different studies, making it more easy for the reader to compare (eg instead of "vit D was measured" for one study, and "insufficiency 25 OH vit D level < 30 ng/ml" for another one, please indicate whether 25 OH vit D measurement was included and which were the results // Same for method for evaluation of bone health, which results).
We thank you for this comment. We have modified table 2, making a separate column for vitamin D results, as you suggested.
line 209: "Dependent on according to"
This line has been changed.
The discussion needs more structure. Try to summarize the results (what is known, what remains unclear) and confront them with results in adults (as is done already in the actual manuscript). How to explain the heterogeneity of the results. Please highlight the heterogeneity of definitions and of the study methods, the lack of information on the content of PN mixtures and protocols used for administration of TPN, and try to end with conclusions for further primary research. How to standardize definitions and methods for the evaluation of bone health?
Line 292-299 is not clear to me. As no information is included on the content of PN mixtures, the discussion in line 333-348 is not really linked to the results presented. The same holds true when mentioning biphosphonates and Denosumab (line 376-383). Try to limit the discussion to what is presented in the result section, or clearly explain why these items should be included in future research on the topic.
Try to be more specific in the conclusion regarding the kind of research needed.
We thank the reviewer for the comments, we have now modified the discussion considering all the suggestions.

Reviewer 2 Report
The topic chosen by authors is very important, because MBD in IF patients is common, important and dangerous complication, difficult to manage in this condition. So recognition of causes and factors influencing it's course and advancement.
Author Response
Reviewer 2
Comments and Suggestions for Authors
The topic chosen by authors is very important, because MBD in IF patients is common, important and dangerous complication, difficult to manage in this condition. So recognition of causes and factors influencing it's course and advancement.
We thank the reviewer for the kind and positive comment.

Round 2
Reviewer 1 Report
The authors have adequately answered to the comments, raised in the previous report, the manuscript has been clearly improved with the results presented as suggested. I do not have further comments.